# Plasma Membrane Blebbing Is Controlled by Subcellular Distribution of Vimentin Intermediate Filaments

**DOI:** 10.3390/cells13010105

**Published:** 2024-01-04

**Authors:** Aleksandra S. Chikina, Anna O. Zholudeva, Maria E. Lomakina, Igor I. Kireev, Alexander A. Dayal, Alexander A. Minin, Mathieu Maurin, Tatyana M. Svitkina, Antonina Y. Alexandrova

**Affiliations:** 1N.N. Blokhin National Medical Research Center of Oncology, 24 Kashirskoe Shosse, Moscow 115478, Russia; aleksandra.chikina@pasteur.fr (A.S.C.); annazholudeva@bk.ru (A.O.Z.); maria.lomakina@mail.ru (M.E.L.); 2Dynamics of Immune Responses Team, INSERM-U1223 Institut Pasteur, 25-28 Rue du Dr Roux, 75015 Paris, France; 3Department of Biology and A.N. Belozersky Institute of Physico-Chemical Biology, M.V. Lomonosov Moscow State University, 1 Leninskie Gory, Moscow 119992, Russia; iikireev@gmail.com; 4Institute of Protein Research, Department of Cell Biology, Russian Academy of Sciences, Moscow 119988, Russia; alexrhea9999@gmail.com (A.A.D.); alexminin@gmail.com (A.A.M.); 5Institut Curie, PSL Research University, INSERM U932, 26 rue d’Ulm, 75248 Paris, France; mathieu.maurin@curie.fr; 6Department of Biology, University of Pennsylvania, Philadelphia, PA 19104, USA

**Keywords:** vimentin intermediate filaments, cell cortex, blebbing, mesenchymal-to-amoeboid transition

## Abstract

The formation of specific cellular protrusions, plasma membrane blebs, underlies the amoeboid mode of cell motility, which is characteristic for free-living amoebae and leukocytes, and can also be adopted by stem and tumor cells to bypass unfavorable migration conditions and thus facilitate their long-distance migration. Not all cells are equally prone to bleb formation. We have previously shown that membrane blebbing can be experimentally induced in a subset of HT1080 fibrosarcoma cells, whereas other cells in the same culture under the same conditions retain non-blebbing mesenchymal morphology. Here we show that this heterogeneity is associated with the distribution of vimentin intermediate filaments (VIFs). Using different approaches to alter the VIF organization, we show that blebbing activity is biased toward cell edges lacking abundant VIFs, whereas the VIF-rich regions of the cell periphery exhibit low blebbing activity. This pattern is observed both in interphase fibroblasts, with and without experimentally induced blebbing, and during mitosis-associated blebbing. Moreover, the downregulation of vimentin expression or displacement of VIFs away from the cell periphery promotes blebbing even in cells resistant to bleb-inducing treatments. Thus, we reveal a new important function of VIFs in cell physiology that involves the regulation of non-apoptotic blebbing essential for amoeboid cell migration and mitosis.

## 1. Introduction

Plasma membrane blebs are special membrane protrusions that are expelled by an increase in intercellular hydrostatic pressure [1,2] and function in different cellular processes, such as cytokinesis [3,4,5], apoptosis [6], and cell rounding after de-adhesion [7]. Blebs also function as protrusive organelles during a particular mode of amoeboid cell motility [2,8], which is characterized by a high speed and independence from the adhesion to the extracellular matrix (ECM) (reviewed in [9]) and the activity of matrix metalloproteases [10,11,12]. Bleb-based motility is typical for free-living amoebae, such as *Dictyostellium*, and for immune cells [13,14,15]. An interest in amoeboid motility has been greatly stimulated by the findings that some tumor and stem cells can switch to amoeboid motility in order to bypass unfavorable conditions for long distance migration and dissemination [2,16,17,18,19].

The transition from lamellipodia-driven mesenchymal motility to bleb-driven amoeboid motility can be experimentally triggered in tumor cells using different approaches, such as by disrupting cell–substrate adhesions [19,20,21,22,23,24], preventing the degradation of the ECM by the inhibition of metalloproteinases [16,19], inhibiting Arp2/3-complex-dependent actin polymerization in lamellipodia [20,25,26,27,28,29], and shifting the balance between actin-polymerization-driven protrusion and actomyosin-dependent contractility toward higher contractility [11,12,20,30,31]. The transition to amoeboid motility can also be stimulated by specific a topology or stiffness of the ECM, and by confinement conditions [32,33,34]. However, the efficiency of this transition varies significantly among cell types and even between individual cells in the same cell population. Thus, different triggers can switch some cells to blebbing while others do not demonstrate such a transition [23,26,27,35]. What determines the differential ability of cells to switch to bleb-driven motility remains poorly understood.

One possibility is that some intrinsic differences between individual cells in the population, most likely in their cytoskeletal organization, explain the heterogeneous blebbing response to bleb-inducing conditions. Actin filaments, microtubules, and intermediate filaments (IFs) are the main components of the cytoskeleton, which play both distinct and cooperative roles in cell motility. Actin filaments are mainly responsible for force-generating activities, such as protrusion, contraction, and adhesion, whereas microtubules play important roles in intracellular trafficking and cell polarity. IFs are mainly involved in regulating mechanical properties of the cell, tissue integrity, and intracellular organization, but also have some signaling functions and can regulate cell–matrix and cell–cell adhesion [36].

In this study, we investigated the mechanistic basis of the heterogeneity exhibited by HT1080 fibrosarcoma cells during their transition to amoeboid motility after pharmacological inhibition of the Arp2/3 complex with CK-666 [23,26,27,35]. We found no substantial differences in the distribution of either actin filaments or microtubules between cells that formed blebs in such conditions and those that did not. However, we found that the distribution of vimentin intermediate filaments (VIFs) was functionally linked to the ability of cells to exhibit blebbing. Specifically, the close proximity of VIFs to the cell periphery protected the adjacent cell edges from either intrinsic or induced plasma membrane blebbing. Thus, we revealed a new important function of IFs in cell physiology—an ability to negatively regulate bleb formation. Such regulation can be very important for the vital cellular processes, such as amoeboid migration and mitosis, where the formation of non-apoptotic blebs plays an essential role. These findings add a new aspect to the growing appreciation that IFs play much more diverse roles in cell physiology than previously thought [36,37,38,39,40].

## 2. Materials and Methods

### 2.1. Cells

HT1080 human fibrosarcoma cells were obtained from ATCC (Manassas, VA, USA); 1036 non-transformed subcutaneous human fibroblasts were kindly provided by Elena Nadezhdina (Institute of Molecular Biology, Moscow, Russia); the MFT-6 (vimentin positive) and MFT-16 (vimentin-null) fibroblast cell lines were generated from wild-type [41] and vimentin knockout mice, respectively [42], and were kindly provided by Robert Evans (University of Colorado, Denver, CO, USA). The MFT-16-HVim cell line with restored vimentin expression was produced using transfection with human vimentin cDNA in the retroviral vector pBABE-puro (Cell Biolabs, San Diego, CA, USA) according to the manufacturer’s instructions as described earlier [43]. Briefly, the virus was produced by co-infecting equal amounts of the pBABE-puro-Vim construct and pCL-Eco helper plasmid (Imgenex, San Diego, CA, USA) into HEK293 cells using TransIT-LT1 Transfection Reagent. Culture supernatant was collected on the second day post-transfection and incubated with the target cells for 4–8 h in the presence of 8 μg/mL polybrene (Sigma, St. Louis, MO, USA). Two days after infection, cells were placed under 2 μg/mL puromycin selection. To avoid the effect of the drug resistance mediated by P-glycoprotein during selection, 2.2 μM of verapamil was added to the culture medium.

To obtain HT1080 cells expressing human EGFP-vimentin (EGFP-vimHT1080), HT1080 cells were transfected with pEGFP-Vimentin-hum [44] kindly provided by Robert Goldman (Northwestern University). For the generation of vimentin knockdown (HT1080shVim) and control (HT1080scramble) cells, the vectors encoding EGFP reporter and either vimentin-targeting (5′-GTACGTCAGCAATATGAAA-3′) or scrambled (5′-ATGTACTGCGCGTGGAGA-3′) shRNA [45] were kindly provided by Robert Goldman (Northwestern University). Cells were transfected using Lipofectamine 2000 (Invitrogen, ThermoFisher Scientific, Waltham, MA, USA) according to the One Tube Protocol by Invitrogen.

For the stable expression of vimentin shRNA in HT1080 cells, we used the pSilencer 5.1 H1 (Clontech, Mountain View, CA, USA) retroviral vector with the inserted sequence encoding vimentin-targeting shRNA [46]. The scrambled sequence in the same vector was used as a control. Two days after infection, cells were placed under 2 mg/mL puromycin selection. The efficiency of knockdown by this vimentin-targeting shRNA was confirmed previously [47]. We confirmed the vimentin silencing by immunoblotting and immunofluorescence.

Cells were cultured in Dulbecco’s modified Eagle’s medium (DMEM) containing 10% fetal bovine serum (FBS) (HyClone, Logan, UT, USA) and 100 U/mL penicillin/streptomycin (Quality Biological, Gaithersburg, MD, USA), and maintained in a humidified atmosphere at 37 °C and 5% CO_2_.

### 2.2. Antibodies and Reagents 

For fluorescence staining, we used the following primary antibodies: mouse monoclonal anti-vimentin V9 (cat. #V6630, Sigma, St. Louis, MO, USA); rabbit polyclonal anti-vimentin D21H3 (cat. #5741, Cell Signaling Technology, Danvers, MA, USA); mouse monoclonal anti-α-tubulin DM1A (MABT205, Sigma-Aldrich, St. Louis, MO, USA); secondary Alexa Fluor594-conjugated Goat Anti-Rabbit IgG (H+L) antibody (Jackson Immunoresearch, Ely, UK); secondary Alexa Fluor 488-conjugated Goat Anti-Mouse IgG antibody (Jackson Immunoresearch); phalloidin-Alexa-Fluor 488 (Thermo Fisher Scientific, Waltham, MA, USA) (dilutions of a stock were prepared according to the manufacturer’s recommendation) and DAPI (Sigma, St. Louis, MO, USA). For fluorescence staining, cells were fixed with 3.7% formaldehyde in PBS at 37 °C for 10 min, washed with PBS three times, and permeabilized with 0.1% Triton X-100 in PBS for 2 min before immunostaining.

For Western blot analysis, the following antibodies were used: rabbit polyclonal anti-vimentin D21H3 (Cell Signalling); mouse monoclonal anti-β-actin, clone C4 (Santa Crus, Santa Cruz, CA, USA); rabbit polyclonal anti-Myosin Light Chain 2 (clone D18E2) (Cell Signalling); rabbit polyclonal anti-P-Myosin Light Chain 2 (Ser19) (clone 3671) (Cell Signalling); anti-γ-tubulin GTU88 (Sigma). Horseradish-peroxidase-conjugated goat polyclonal anti-mouse and anti-rabbit IgG antibodies (Jackson Immunoresearch, Ely, UK) were used as secondary antibodies. All antibodies and dyes were used in the concentrations recommended by their product specification sheets.

CK-666 (TOCRIS Bioscience, Abingdon, UK) was used at the final concentration of 100 and 200 µM prepared from 50 mM solution in DMSO. Colcemid (Sigma) was used at the final concentration of 0.2 µM in culture media prepared from 1 mg/mL stock solution in DMSO. Withaferin A (WFA) (CAS 5119-48-2-Calbiochem, Burlington, MA, USA) was used at the final concentration of 1 μM prepared from 1 mM stock solution in DMSO.

### 2.3. Western Blot Analysis

Cells were washed twice with wash buffer (10 mM Tris-HCl pH7.5, 0.5 mM EDTA, 150 mM NaCl) and lysed with lysis buffer (wash buffer supplemented with 0.5% Na deoxycholate, 1% NP-40, protease inhibitor cocktail (La Roche Ltd., Basel, Switzerland), and phosphatase inhibitor cocktail (Sigma-Aldrich, USA). Samples were mixed with 5× sample buffer (250 mM Tris-HCl pH 6.8, 10% SDS, 30% glycerol, 5% β-mercaptoethanol, 0.02% bromophenol blue), heated for 10 min at 95 °C, and loaded onto SDS-polyacrylamide gel in equal protein concentrations according to the SDS-PAGE Bio-Rad protocol. Resolved proteins were transferred to Amersham Hybond-C-nitrocellulose hybridization membranes, 0.45 micron (GE Healthcare, Chicago, IL, USA). Membranes were blocked with 5% m/v bovine serum albumin solution (PanEco, Moscow, Russia) in Tris-buffered saline with 0.1% *v*/*v* of Tween 20 (CAS 9005-64-5, AppliChem, Darmstadt, Germany) for 1 h followed by incubation with the primary antibodies at 4 °C overnight. After washing, peroxidase-conjugated secondary antibodies were applied for 1 h at room temperature. Blotted protein bands were detected using Pierce ECL Western Blotting Substrate (ThermoFisher Scientific, Waltham, MA, USA), and chemiluminescence images were captured by Image Quant LAS4000 (GE Healthcare, Chicago, IL, USA). The densitometry of results was performed using ImageJ software, version 1.53c, and the results are presented as means with standard errors of the mean (SEM) from three independent experiments. 

### 2.4. Light Microscopy

DIC and epifluorescence microscopy was performed using Nikon Eclipse Ti-E microscope equipped with Plan Fluor 40×/1.30 Oil objective, Hamamatsu ORCA-ERC 4742-80 camera (Hamamatsu Photonics, Hamamatsu, Japan), and Nis-Elements AR 4.11.0 software (Nikon, Tokyo, Japan) and also using Leica TCS SP5 11 confocal laser scanning microscope described below. Phase contrast microscopy for correlative PREM analyses was performed using Eclipse TE2000-U inverted microscope (Nikon USA, Melville, NY, USA) equipped with Plan Apo 100×/1.3 NA objective and Cascade 512B CCD camera (Photometrics, Tucson, AZ, USA) driven by MetaMorph software, version 7.10.4 (Molecular Devices, Sunnyvale, CA, USA). Confocal microscopy was performed using Leica TCS SP5 11 confocal laser scanning microscope with HCX PL APO CS 20 × 0.7 or HDX PL APO 63 × 1.4 objectives controlled by LAS-AF v. 2.7.3.9723 software (Leica Microsystems, Wetzlar, Germany). Structured illumination microscopy (SIM) was performed using Nikon N-SIM microscope with 100×/1.49 NA oil immersion lens using 488 nm and 561 nm diode laser excitation. Series of focal planes with 120 nm steps were acquired in 3D-SIM mode with EMCCD camera iXon 897 (Andor, Belfast, Northern Ireland); an effective pixel size was adjusted to 60 nm with 2.5 × tube lens. Exposure conditions were adjusted to obtain a typical yield of about 5000 max counts (16-bit raw image). Image acquisition for SIM reconstruction were performed using NIS-Elements AR 5.2 software (Nikon, Melville, NY, USA).

To characterize the changes in cell protrusion activity under drug treatment, cells were plated onto glass-bottom dishes, cultivated for 24 h, and imaged in DIC. Then, they were treated with CK-666 (200 μM, 1 h), colcemid (0.2 μM, 1 h), Withaferin A (WFA, 1 μM, 3 h), or an appropriate vehicle, and imaged again. It was important to image living cells because fixation could destroy some blebs. The number of mesenchymal and blebbing cells and cells with intermediate morphology (in the case of MFT6, MFT-16, and MFT-16HVim cells and for WFA treatment) was counted manually on these images before and after the treatment and expressed as a fraction of blebbing cells in each field of view. The obtained fractions were statistically evaluated for each condition. WFA-treated cells were also fixed for subsequent immunofluorescence staining.

For correlative DIC and immunofluorescence microscopy, cells plated onto etched coverslips (#1916-91818, Bellco Glass, Vineland, NJ, USA) were treated with 200 µM CK-666 for 1 h and imaged by time-lapse DIC microscopy to record locations of blebbing and non-blebbing cells. Then, cells were either fixed or CK-666 was washed out, and after the cells restored mesenchymal morphology, they were fixed and processed for immunofluorescence, as described above. The imaged cells were identified using an etched grid and imaged by epifluorescence microscopy or SIM.

For the analysis of the association of the blebbing ability with the initial distribution of VIFs in untreated cells, HT1080 cells transfected with EGFP-vimentin were plated onto glass-bottom dishes and after 24 h in culture were imaged by time-lapse DIC and immunofluorescence microscopy for 30 min with 1 min intervals. Next, these cells were treated with 200 µM CK-666 and imaged by time-lapse microscopy for an additional hour. The fraction of the cell area occupied by VIFs before CK-666 treatment was determined using the DIC and EGFP-vimentin images of the cells and correlated with the protrusive behavior of the same cells (blebbing or non-blebbing) following the CK-666 treatment.

### 2.5. Correlative Platinum Replica Electron Microscopy (PREM)

Detailed procedures for correlative PREM were described previously [48]. Briefly, cells were grown on homemade glass-bottom dishes with a gold-shadowed locator pattern, treated with 200 µM CK-666 for 1 h, imaged by phase contrast microscopy to identify blebbing and non-blebbing cells, washed from CK-666 and, after the cells restored mesenchymal morphology, extracted by adding an excess of the extraction solution to the dish on a microscope stage. The extraction solution contained 1% Triton X-100 in PEM buffer (100 mM PIPES-KOH [pH 6.9], 1 mM MgCl2, 1 mM EGTA), and 2 µM unlabeled phalloidin (for samples not treated with gelsolin) or 10 µM taxol (for samples treated with gelsolin). After extraction at room temperature for 2–3 min and three quick rinses with PEM buffer containing 2 µM unlabeled phalloidin or 10 µM taxol, samples were fixed with 2% glutaraldehyde in 0.1 M sodium cacodylate buffer (pH 7.3) for 20 min, either immediately after washing or after additional treatment of extracted cells with gelsolin according to the previously described protocol [49,50]. Incubation with primary rabbit antibody against non-muscle and smooth-muscle myosin heavy chains (#BT-564; Biomedical Technologies; diluted 1:20 in PEM buffer) was performed on unfixed gelsolin-treated samples. After that, samples were fixed with glutaraldehyde, quenched with NaBH4, and stained with donkey anti-goat secondary antibody conjugated with 18 nm colloidal gold (#705-205-147; Jackson ImmunoResearch, 1:5 dilution) as described previously [48]. After additional glutaraldehyde fixation, samples were sequentially treated with aqueous tannic acid (1 mg/mL) and uranyl acetate (2 mg/mL), critical point dried, rotary shadowed with platinum at ~45° angle and carbon at ~90° angle, transferred onto electron microscopic grids, and analyzed using JEM 1011 transmission electron microscope (JEOL USA, Peabody, MA, USA) operated at 100 kV. Images were captured with ORIUS 832.10W CCD camera (Gatan, Pleasanton, CA, USA) and presented in inverted contrast. Light microscopy and PREM images were aligned using Adobe Photoshop. Pseudocolor to PREM images was applied using the Hue/Saturation tool in Photoshop for selected areas.

### 2.6. Image Analysis

All images were processed using ImageJ/FIJI (NIH) and Adobe Photoshop CS2 or 24.7.0 software.

To determine VIF density in PREM images of gelsolin-treated mesenchymal and blebbing cells, we selected cell regions adjacent to the leading edge with a depth (from the leading edge into the cell interior) of 3 μm and widths along the leading edge ranging from 1.7 μm to 4.2 μm. The total length of VIFs within these regions was normalized to the region area.

To analyze the level of vimentin expression and its depletion, we measured the fluorescence intensity of immunofluorescence images of control (HT1080scramble) and HT1080shVim cells. For this purpose, cell contours were outlined on the phalloidin-stained image and transferred to the vimentin-stained image; then, the intensities of VIF fluorescence per cells were measured. The graph was plotted using the average intensities of 30 cells.

The percentage of the cell area occupied by vimentin in EGFP-vimentin-expressing HT1080 cells was determined using ImageJ by measuring the cell area occupied by EGFP-vimentin in fluorescence images and comparing it to the total cell area in the corresponding DIC images.

To analyze the radial distribution of VIF staining in HT1080 and 1036 cells, we defined ten concentric regions for each cell from the cell center to the cell border using ImageJ/FIJI. For this purpose, we first applied a threshold to the F-actin channel to obtain a binary mask of the cell. Then, we created a distance map, which defined the distance of each pixel of the mask to the nearest cell border. The maximum distance on the map (corresponding to the cell center) was determined from the distance map. Next, the cell mask was divided into ten regions by eroding the mask by 1/10th of its maximum distance at each step to create ten circumferential zones. The most peripheral zone of each cell was discarded, and mean fluorescence intensity of vimentin immunostaining was measured for each of the nine remaining zones to obtain a radial distribution profile. Radial profiles for all cells were then normalized to obtain the same area under the curve. Custom ImageJ macros for this analysis are provided in Appendix A.

### 2.7. Statistical Analysis

Statistical analyses were performed using GraphPad Prism version 8.1.1 or 10.1.0 for Windows (GraphPad Software, Boston, MA, USA) and Excel software, version 16.43 (Microsoft). The details for the analysis of each type of experiment are presented in the corresponding figure legends.

## 3. Results

### 3.1. Distribution of Intermediate Filaments Differentiates Cells That Exhibit or Do Not Exhibit Induced Blebbing

The population of HT1080 fibrosarcoma cells cultured on a glass surface contained very few spontaneously blebbing cells (between 0 and 5% depending on the experiment), whereas all other cells had normal mesenchymal morphology with lamellipodia and ruffles at the cell periphery. After treatment with the Arp2/3 complex inhibitor CK-666 (200 µM for 1 h), all HT1080 cells lost their typical mesenchymal protrusions, lamellipodia, and ruffles, but only about 30% of cells exhibited blebbing [27]. This transition was reversible, and lamellipodia were restored after CK-666 withdrawal in both cell subpopulations (Figure 1A,D and Appendix A).

To evaluate our hypothesis that the intrinsic differences between individual cells controlling their ability to form blebs result from their distinct cytoskeletal organization, we analyzed high-resolution cytoskeleton architecture using correlative platinum replica electron microscopy (PREM) [48,51]. Specifically, we imaged live CK-666-treated cells by phase contrast microscopy to identify blebbing and non-blebbing cells. Then, we washed out CK-666, allowed cells to restore lamellipodia, and prepared them for PREM. We initially used a conventional detergent extraction procedure, which preserves all cytoskeleton components. Because actin filaments are very abundant in HT1080 cells, this approach mostly revealed the organization of the actin cytoskeleton, whereas other cytoskeleton components—IFs, non-muscle myosin II (NMII) filaments, and microtubules—were largely masked by dense actin filaments. Using this procedure, we were unable to find any striking differences in the actin cytoskeleton organization between cells that formed blebs under CK-666 treatment and those that did not (Appendix A).

We therefore considered the possibility that important cytoskeletal differences defining the blebbing behavior could be exhibited by non-actin cytoskeleton components. Therefore, we treated detergent-extracted cells with the actin-severing protein gelsolin to remove actin filaments and thus expose microtubules, NMII filaments, and IFs [52] (Figure 1). In PREM samples, microtubules and IFs can be distinguished by their thicknesses, ~25 nm and ~14 nm, respectively, including the platinum layer (Figure 1C,F). To better visualize NMII filaments, we used immunogold staining. We did not detect striking differences in the distribution of microtubules or NMII in the actin-depleted cytoskeletons derived from the cells that produced blebs under CK-666 treatment and from those that did not. The distribution of NMII was highly variable within each of these cell categories, but it was not obviously different between them. On the other hand, the distribution of IFs (highlighted in blue in Figure 1C,F) was significantly different between these two cell categories. In cells that did not form blebs under CK-666 treatment (Figure 1D–F), numerous IFs could be seen in close proximity to the cell edge, where they were distributed among microtubules (Figure 1F). In contrast, IFs were very scarce in the peripheral regions of HT1080 cells that exhibited blebbing prior to CK-666 washout (Figure 1A–C). These differences were confirmed by quantification of IF density in peripheral cell regions of non-blebbing and blebbing cells (Appendix A). These data reveal a possible correlation between the IF distribution and cell blebbing and suggest that the IF distribution can be an important factor controlling the cell’s competence to exhibit blebbing. In subsequent experiments, we focused on IFs in order to test this hypothesis.

### 3.2. Subcellular Distribution of VIFs Correlates with Both Spontaneous and CK-666-Induced Cell Blebbing

We next used light microscopy to correlate the subcellular IF distribution and blebbing activity of cells on a broader scale. Given the fibroblastic origin of HT1080 cells, vimentin is the main component of their IFs. Accordingly, we used immunofluorescence staining of vimentin in CK-666-treated HT1080 cells and analyzed the vimentin distribution by confocal microscopy (Figure 2A,B). The total immunofluorescence intensity of vimentin per cell was similar in the cells that formed blebs in the presence of CK-666 (0.0845 ± 0.0413 arbitrary units, mean ± SD) and those that did not (0.0956 ± 0.0481; *p* = 0.4; n = 19 blebbing cells and 24 non-blebbing cells). However, in non-blebbing cells, the VIF network was distributed throughout the cell extending from the perinuclear area up to the cell periphery (Figure 2A). In contrast, VIFs in blebbing cells were mainly concentrated in the perinuclear region and only sparsely populated the cell periphery (Figure 2B). Accordingly, quantitative analysis revealed significant differences in the subcellular distribution of VIFs between these two cell categories (Figure 2C, red and blue traces). Specifically, the intensity of vimentin immunostaining in blebbing cells was more concentrated in the perinuclear region and relatively depleted at the cell periphery, as compared with the cells without blebs.

To determine whether the correlation between blebbing behavior and VIF distribution also holds true for spontaneously blebbing cells, we performed correlative DIC imaging of live untreated HT1080 cells, which was followed by post hoc vimentin immunostaining of the same cells and their analysis by a super-resolution fluorescence microscopy technique—structured illumination microscopy (SIM). Untreated HT1080 cells exhibited a small fraction (0–5% depending on the experiment) of spontaneously blebbing cells. We found that VIFs in these spontaneously blebbing HT1080 cells were concentrated in the perinuclear region (Figure 2D), similar to cells in which blebbing was induced by the CK-666 treatment. Among the untreated HT1080 cells with lamellipodia and ruffles, but without blebs, most cells contained well-spread VIF networks that reached the cell periphery (Figure 2E). Only 6.7% of these non-blebbing cells exhibited perinuclear accumulation of VIFs with sparse distribution at the cell periphery (2 cells from a total of 30 cells from 3 experiments).

We also analyzed the VIF distribution in normal subcutaneous 1036 fibroblasts, which did not switch to blebbing even under CK-666 treatment [27]. We found that these cells contained well-distributed VIFs that closely approached the cell periphery (Appendix A), as was confirmed by quantification of the VIF distribution in these cells (Figure 2C, green trace). Thus, a lack of blebbing behavior in these cells also correlates with their broad intracellular VIF distribution.

### 3.3. Asymmetric Subcellular Distribution of VIFs Correlates with Biased Blebbing Activity at the Cell Edges

We next asked whether an asymmetric VIF distribution in CK-666-treated cells correlated with differential blebbing activity at the cell edges of these cells (Figure 3A–C). To address this question, we expressed EGFP-vimentin in HT1080 cells (EGFP-vimHT1080) and observed both the VIF distribution and blebbing activity in individual cells by live-cell imaging. We found that the cells, in which EGFP-vimentin was nearly symmetrically concentrated in the perinuclear area and did not closely approach the cell edges, typically formed blebs all around the cell perimeter under CK-666 treatment (Figure 3A). In contrast, in cells with polarized blebbing, VIFs were located at a significant distance from the actively blebbing edge but were found in close proximity to the non-blebbing cell edge (Figure 3B). To better visualize the intracellular VIF distribution during blebbing, we used correlative SIM, in which live cells were video recorded by DIC microscopy in the presence of CK-666, fixed, double-stained with vimentin antibody and phalloidin, and analyzed by SIM (Figure 3C–E). We found that in all pre-recorded cells, high blebbing activity occurred at the cell edges that were relatively depleted of VIFs (Figure 3E, green dashed line). Conversely, the enrichment of VIFs at the cell periphery correlated with low or no blebbing activity at these regions (Figure 3E, yellow dashed line).

These results showed a consistent correlation between blebbing activity and the low abundance of VIFs at the cell periphery, not only among individual cells, but also at a subcellular level within individual cells.

### 3.4. Initial Distribution of VIFs Predicts Blebbing Cell Behavior after CK-666 Application

Our data so far suggest that VIFs located at the cell periphery might be able to locally suppress cell blebbing. Alternatively, the perinuclear VIF distribution could be a consequence of blebbing activity, as if VIFs were displaced from the cell periphery by retracting blebs. To distinguish between these possibilities, we compared the distribution of VIFs prior to the induction of blebbing with the subsequent cell response.

For this purpose, we imaged untreated EGFP-vimHT1080 cells by both DIC and fluorescence live-cell microscopy, then added CK-666 and observed the resulting changes in protrusive cell behavior over time (Figure 3F–J). We found that the cells, which switched to blebbing under CK-666 treatment (Figure 3H,I), before CK-666 application had VIFs mainly collapsed in the perinuclear region (Figure 3H,J). In contrast, the pre-existing VIF network was broadly distributed in cells, which retracted under CK-666 treatment but did not form blebs (Figure 3F,G,J). These results suggest that the ability of cells to switch to blebbing correlates with the pre-existing low abundance of VIFs at the cell periphery.

### 3.5. Blebbing Activity in Mitotic Cells Correlates with Low Abundance of VIFs at the Cell Periphery

Having established a robust negative correlation between the VIF peripheral enrichment and blebbing activity in motile interphase cells, we asked whether the same rule applies to membrane blebbing associated with cytokinesis. Using EGFP-vimHT1080 cells, we simultaneously monitored blebbing activity and VIF dynamics in dividing cells by DIC and fluorescence microscopy, respectively (Figure 4). During early cytokinesis, when cells had just begun to re-spread after mitosis, they exhibited active membrane blebbing at the polar regions of both daughter cells. At this stage, VIFs were aggregated in the perinuclear area close to the cleavage furrow, but away from the blebbing edges (Figure 4A). As cytokinesis progressed, and the cells continued to spread, the VIF networks in daughter cells gradually redistributed from the perinuclear area toward the polar regions. This redistribution could occur synchronously in both daughter cells, or faster in one cell than in the other. In the case of asynchronous redistribution of VIFs during late cytokinesis (Figure 4B), the daughter cells, in which VIFs distributed throughout the cytoplasm to a greater extent, ceased blebbing and formed lamellipodia sooner than their sister cells with aggregated VIFs, which continued to bleb until their VIFs also redistributed and reached the distant cell periphery (Figure 4C).

Thus, our results show that plasma membrane blebbing in dividing cells, similar to blebbing in interphase cells, correlates with, and possibly is controlled by, the distribution of VIFs.

### 3.6. Enforced Removal of VIFs from the Cell Periphery Promotes Blebbing

The observed strong correlation between low abundance of VIFs at the cell periphery and enhanced blebbing activity at the same edges raises a question of whether there is a causative relationship between these two phenomena. To address this question, we pharmacologically displaced VIFs from the cell periphery. It has been known for decades that the cytoplasmic distribution of VIFs in cells depends on the presence of microtubules, whereas microtubule depolymerization leads to a collapse of VIFs in the perinuclear area [53]. Accordingly, the treatment of HT1080 cells with the microtubule-depolymerizing drug colcemid (0.2 µM for 1 h) resulted in a collapse of VIFs in the perinuclear area (Appendix A). Strikingly, the majority of colcemid-treated HT1080 cells exhibited active blebbing all over the cell periphery even in the absence of CK-666 treatment (Figure 5A top, Appendix A). After colcemid washout, cells recovered lamellipodia and stopped blebbing (Appendix A).

Given this striking effect of colcemid on HT1080 cells, we then tested the response to colcemid of 1036 fibroblasts, which normally are resistant to the CK-666-induced blebbing [27]. As expected, colcemid treatment of 1036 cells efficiently induced the VIF collapse to the cell center (Appendix A). Remarkably, the majority of colcemid-treated 1036 cells exhibited active blebbing around the cell perimeter in the absence of CK-666, although the blebs were smaller than in HT1080 cells (Figure 5A bottom, Appendix A). A few colcemid-treated 1036 cells that did not form blebs nonetheless exhibited peripheral swellings, which resembled analogous swellings in HT1080 cells that were about to start blebbing [26] (Appendix A).

Taking into account that colcemid treatment leads to disruption of microtubules, in addition to collapsing VIFs, we looked for another approach to deplete VIFs from the cell periphery without the disruption of microtubules. For this purpose, we used Withaferin A (WFA), a steroidal lactone from the plant *Withania somnifera*, which was shown to bind vimentin in vitro and cause the reorganization of VIFs into perinuclear aggregates in cells [54]. We treated HT1080 cells with 1 μM WFA for 3 h and found that this treatment led to the significant accumulation of VIFs in the perinuclear region, whereas the microtubule distribution was not changed (Figure 5B, left). Importantly, WFA treatment also led to a significant increase in the fraction of blebbing cells in the cell population even without additional stimuli (Figure 5B, right).

### 3.7. Genetic Vimentin Depletion Stimulates Plasma Membrane Blebbing

We next evaluated the roles of VIFs in blebbing by knocking down vimentin from HT1080 cells using shRNA (Figure 6A–C, Appendix A). In one approach (Figure 6A–C), we delivered vimentin shRNA to HT1080 cells using a retroviral vector, which yielded a stable vimentin knockdown cell line (HT1080shVim cells). Western blot analysis demonstrated about 65% depletion of vimentin in HT1080shVim cells (Figure 6A). Immunostaining of HT1080shVim cells also showed a significant reduction of vimentin levels by ~70% in these cells (Figure 6B). The remaining VIFs were concentrated in the perinuclear region and did not extend to the cell periphery (Figure 6B, lower image). When we compared the ability of HT1080 and HT1080shVim cells to exhibit blebbing under CK-666 treatment, we found that almost 60% of HT1080shVim cells showed blebbing after application of CK-666, as compared with 35% of blebbing cells in the control HT10180scramble culture (Figure 6C).

As a second knockdown approach, we transiently transfected HT1080 cells with a plasmid encoding both vimentin shRNA and GFP (Appendix A). In this case, the transfected cells could be recognized by GFP fluorescence. The HT1080 cells transfected with a control scrambled shRNA (Appendix A) did not show any obvious shape alterations, as compared with untransfected HT1080 cells. In contrast, the expression of vimentin shRNA led to a dramatic increase in the fraction of cells with blebs and/or edge swellings (pre-blebbing state), even in the absence of CK-666 treatment (Appendix A).

To achieve full suppression of vimentin expression, we tried to knock out vimentin in HT1080 cells using the CRISPR/Cas9 approach. However, these attempts were unsuccessful, because HT1080 cells totally lacking vimentin could not properly attach to the substrate. This outcome is consistent with previous studies showing deficient cell adhesion after partial vimentin knockdown [55,56]. Given that focal adhesions of HT1080 cells are already very weak and mostly represented by small focal complexes, it is reasonable to expect that complete vimentin ablation could lead to a total cell adhesion failure. Therefore, we instead used the vimentin-null MFT-16 fibroblast line obtained from vimentin knockout mice and compared these cells to the matching control MFT-6 cell line produced from wild-type mice [41,42]. To control for the specificity of the phenotype of vimentin knockout, we generated the MFT-16-HVim cell line by reintroducing human vimentin into vimentin-null MFT16 cells (Figure 6D,E). The cells of all three lines (MFT-6, MFT-16 and MFT-16-HVim) were well-spread on the substrate and predominantly had typical mesenchymal morphology with lamellipodia, but also small fractions of spontaneously blebbing cells and cells with an intermediate phenotype characterized by the presence of peripheral swellings. The percentage of cells with these different phenotypes was similar among all three cell lines treated with DMSO and increased in all of them after treatment with the 100 µM CK-666. However, the increase in the fraction of blebbing cells was significantly greater in MFT-16 cells lacking VIFs in comparison to MFT-6 cells with endogenous vimentin expression. The reintroduction of vimentin into MFT-16 cells led to a decrease in the percentage of blebbing MFT-16-HVim cells to the level exhibited by control MFT-6 cells (Figure 6E). To address the possibility that blebbing regulation in this cell system is a result of some alteration in microtubule organization or in cell contractility we performed immunofluorescence staining of tubulin, as well Western blot analysis of total and active (phosphorylated) Myosin Light Chain (MLC and pMLC, respectively) in these three cultures. We found no apparent differences in microtubule distribution nor in the amount of either MLC or pMLC (Appendix A).

## 4. Discussion

Our results show that the reduction in VIFs at the cell periphery greatly stimulates plasma membrane blebbing, whereas peripheral enrichment of VIFs locally prevents the blebbing response. This conclusion is validated using functional interference with the VIF network in cells by several methods and strengthened by correlative approaches demonstrating that blebbing activity is preferentially expressed in cells with perinuclear distribution of VIFs and is strongly biased toward cell edges with a reduced abundance of VIFs in the cells with asymmetric VIF distribution.

We show that such an association between the blebbing location and the VIF distribution exists in both interphase and mitotic cells. It has long been known that mitosis is naturally accompanied by plasma membrane blebbing [7]. During cell division, blebbing is associated with cell rounding in the prometaphase, but is especially prominent during cytokinesis [3,57], where membrane blebbing can act as valves that locally release cortical contractility and thus stabilize cleavage furrow positioning [5]. On the other hand, vimentin expression and interaction with the actin cortex have been found to be required for successful cell division in confinement [40]. Our results can link these two previously established sets of data about separate roles of blebbing and vimentin in cytokinesis. We propose that VIFs can regulate blebbing by interacting with the cell cortex, while blebbing subsequently ensures proper mitotic division by stabilizing the division plane.

A similar concept can also be applied to cell motility. Thus, it has been shown that VIFs regulate amoeboid motility and cell migration in confined conditions [55,58,59]. On the other hand, these motility modes are known to involve blebbing [4], suggesting again that VIFs can regulate cell motility in confinement by controlling the distribution of blebbing activity. The mechanisms by which VIFs can regulate blebbing ability remain to be fully understood, but several possibilities can be considered. It was shown earlier that plasma membrane blebbing is largely promoted by local weakening of the actin cortex or its detachment from the plasma membrane [2], which can occur either spontaneously, for example at filopodia bases [26,60], or induced experimentally [61,62]. Therefore, it is likely that VIFs are able to strengthen the cortical actin cytoskeleton, thereby protecting the overlaying plasma membrane from blebbing. Consistent with this idea, VIFs in interphase cells can interact with actin filaments directly [63] or through IF-associated proteins, such as plectin [64] or CARMIL2 [65]. Furthermore, VIFs were observed to enmesh with the actin cortex in mitotic cells [66] and regulate actin cortex organization during mitosis in a plectin-dependent manner [40]. Recently it was shown that in mouse embryonic fibroblasts VIFs and F-actin could form an interpenetrating network within cell cortex, which could significantly affect cortex properties [67]. The reported ability of IFs to enhance mechanical properties of the cell [68,69] can rely, in part, on their interactions with the cortical actin cytoskeleton and lead to anti-blebbing protection.

Another possible mechanism by which VIFs can regulate cell blebbing is through signaling-based modulation of cell contractility, which is also known to promote blebbing. Indeed, vimentin knockdown by RNAi or knockout by CRISPR/Cas9 have been recently found to increase RhoA activity in U2OS cells, which in turn activates myosin-II-dependent contractility [70]. This mechanism could be analogous to how the disruption of microtubules leads to an increase in RhoA activity [71,72]. Therefore, stimulation of blebbing by microtubule disruption can be explained by RhoA activation. Multiple studies have shown that blebbing is stimulated by an increase in RhoA activity, which leads to the reassembly of the actin cortex and thus regulates both bleb expansion and retraction [73,74]. On the other hand, we did not find differences in myosin II activity among MFT6, MFT16, and MFT16-HVim cells. Also, neither changes in myosin II activity and traction force generation [59], nor effects on cortical stiffness [75] have been detected in vimentin-null mouse fibroblasts by others. We also did not detect differences in bleb formation between control and vimentin-deficient cells at the basal level, but the differences became clearly apparent after additional stimulation by CK-666. A possible explanation of these phenomena is that cells derived from knockout mice had much more time to compensate for the aberrantly high contractility produced by the downregulation of vimentin, as compared with relatively short-term vimentin depletion in cultured cells, in which the level of blebbing showed an upward trend even at basal level although it did not reach statistical significance. We also used fibroblasts derived from vimentin knockout mice and observed enhanced cell blebbing in cells depleted of VIFs. Moreover, the transfection of vimentin into these cells led to a decrease in the fraction of blebbing cells to the level found in control mouse fibroblasts. Thus, we propose that VIFs can protect cells from blebbing not only through RhoA signaling, but also more directly by interacting with the cell cortex and regulating its architecture.

Regardless of whether VIFs regulate plasma membrane blebbing by directly interacting with the cell cortex or through signaling, our data reveal that VIFs have an important impact on both normal and pathological cell behavior during cell division, motility and morphogenesis.

## Figures and Tables

**Figure 1 cells-13-00105-f001:**
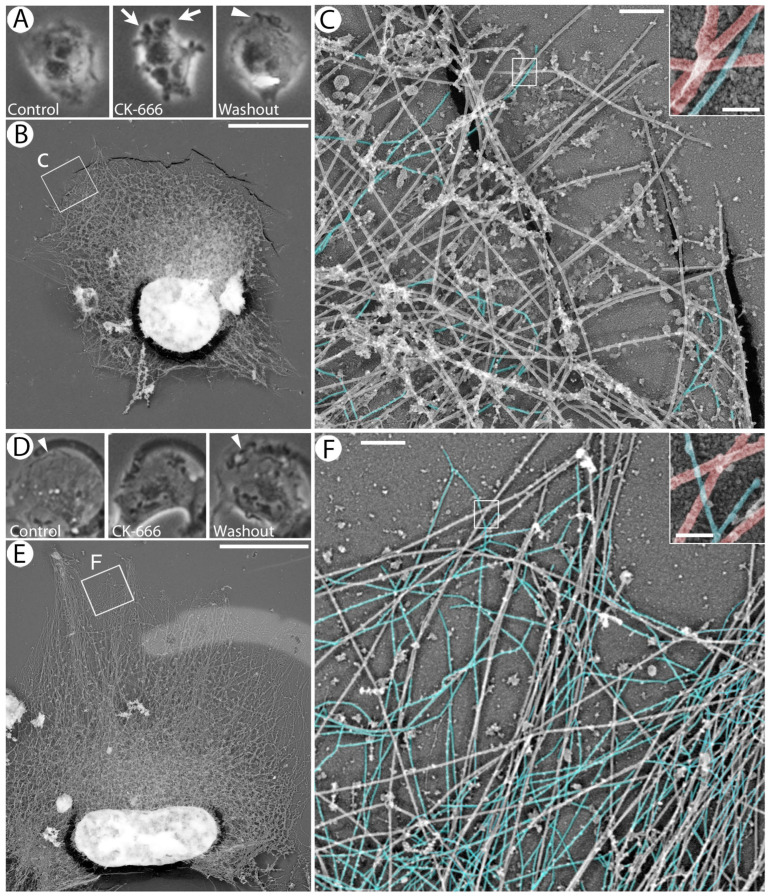
Correlative phase contrast microscopy and PREM of cells treated with 200 µM CK-666 for 1 h and then allowed to restore lamellipodia after drug washout. (**A**–**C**) Example of a cell that switched to blebbing under CK-666 treatment. (**D**–**F**) Example of a cell that did not switch to blebbing under CK-666 treatment. (**A**,**D**) Phase contrast images taken before CK-666 application (**left**), under 1 h CK-666 treatment (**middle**) and 1 h after CK-666 washout (**right**). Blebs and lamellipodia are marked by arrows and arrowheads, respectively. (**B**,**E**) Correlative PREM images with NMII immunogold staining after detergent extraction and gelsolin treatment of the cells shown in (**A**) and (**D**), respectively. (**C**,**F**) Enlarged boxed regions from (**B**) and (**E**), respectively. IFs are pseudocolored in blue; unlabeled long filaments represent microtubules. Boxed regions in main panels are enlarged in insets, where microtubules are pseudocolored in red and IFs in blue. Bars, 10 µm (**A**,**B**,**D**,**E**), 500 nm (**C**,**F**), and 50 nm ((**C**,**F**) insets).

**Figure 2 cells-13-00105-f002:**
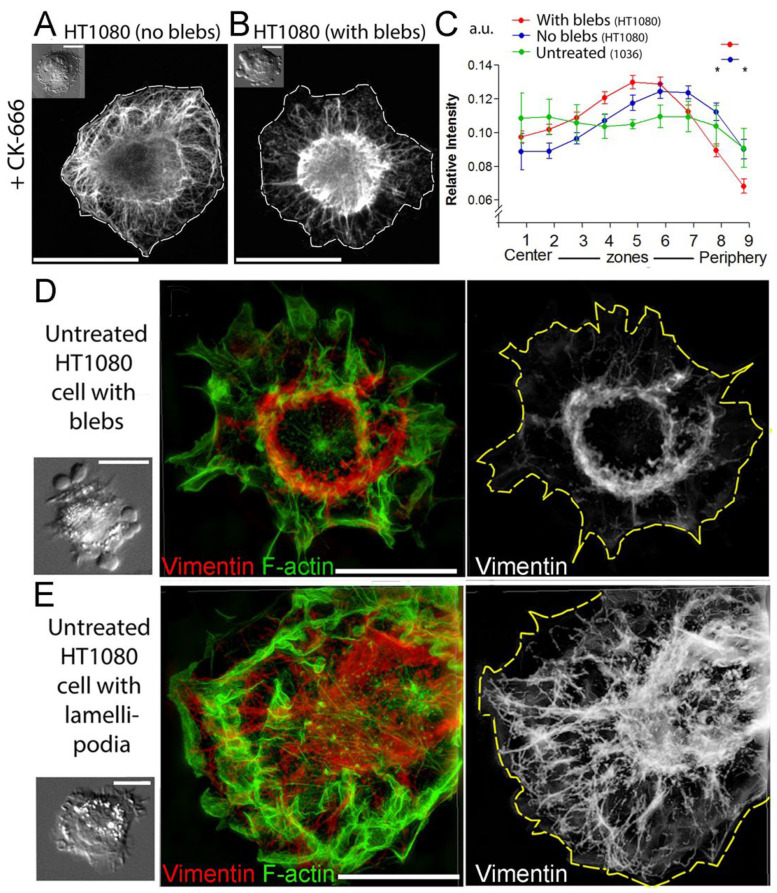
Vimentin enrichment at the cell periphery negatively correlates with cell edge blebbing. (**A**,**B**) Correlative DIC (inserts) and anti-vimentin immunofluorescence microscopy (main panels) of HT1080 cells treated with 200 μM CK-666. (**A**) In the non-blebbing cell, the VIF network is distributed throughout the cell up to the cell periphery. (**B**) In the blebbing cell, the VIF network is sparse at the cell periphery and concentrates in the perinuclear region. (**C**) Center-to-periphery distribution of the mean immunofluorescence intensity of vimentin in blebbing (red, n = 24 cells) and non-blebbing (blue, n = 19 cells) HT1080 cells treated with CK-666 and in untreated normal 1036 fibroblasts (green, n = 7 cells); mean ± SEM; *, *p* < 0.05; unpaired *t*-test with correction for multiple comparisons using Holm–Sidak method comparing data for blebbing and non-blebbing HT1080 cells for each zone. (**D**,**E**) Correlative DIC microscopy (**left panels**) and SIM of untreated blebbing (**D**) and non-blebbing (**E**) HT1080 cells stained with vimentin antibody and phalloidin for F-actin. Cell contours are outlined in yellow. Scale bars: 15 µm (**A**,**B**) and 10 µm (**D**,**E**).

**Figure 3 cells-13-00105-f003:**
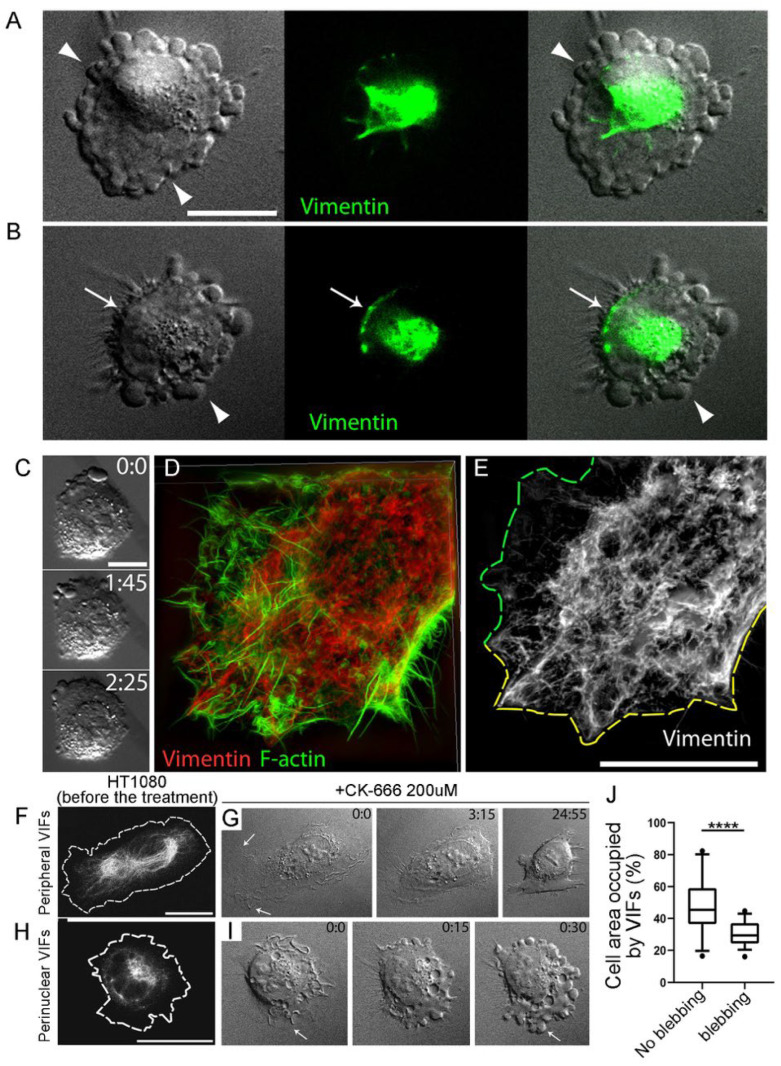
Subcellular distribution of VIFs correlates with and predicts blebbing behavior in HT1080 cells. (**A**,**B**) EGFP-vimHT1080 cells treated with 200 µM CK-666 shown by DIC (**left**) and vimentin immunofluorescence microscopy (**middle**), and as a merged image (**right**). (**A**) Cell exhibiting non-polarized blebbing around the cell perimeter (arrowheads) has a relatively symmetric perinuclear distribution of VIFs with low levels of VIFs at the cell periphery. (**B**) A cell exhibiting polarized blebbing at one side of the cell (arrowhead) has an asymmetric distribution of VIFs enriched at the non-blebbing edge (arrow) and depleted at the blebbing edge (arrowhead). (**C**–**E**) Correlative DIC microscopy ((**C**), frames from time-lapse video), and SIM (**D**,**E**) of the HT1080 cell with asymmetric blebbing induced by treatment with 200 µM CK-666 for 1 h. Blebs form at the cell edge devoid of a dense VIF network ((**E**), green outline), but not at the cell edge enriched in VIFs ((**E**), yellow outline). Vimentin is visualized by immunostaining; actin is stained by phalloidin. (**F**–**J**) Pre-existing VIF distribution predicts blebbing activity after application of CK-666. (**F**,**H**) Distribution of VIFs either throughout the cell (**F**) or in the perinuclear region (**H**) in EGFP-vimHT1080 cells before CK-666 application. (**G**,**I**) Frames from time-lapse DIC sequences of the cells shown in F and H, respectively, after CK-666 application. The cell with widely distributed VIFs (**F**) retracts but does not switch to blebbing for up to ~25 min after addition of 200 µM CK-666 (**G**). The cell with pre-existing perinuclear enrichment of VIFs (**H**) switches to blebbing within 30 sec after addition of 200 µM CK-666 (**I**). Time in min:sec. Scale bar 20 µm. (**J**) Percentage of area occupied by VIFs in cells exhibiting or not exhibiting blebbing after treatment with 200 µM CK-666 for 1 h. Box plot shows the quartiles (box), the 5th and 95th percentiles (whiskers), and outliers (dots); ****, *p* < 0.0001; unpaired *t*-test.

**Figure 4 cells-13-00105-f004:**
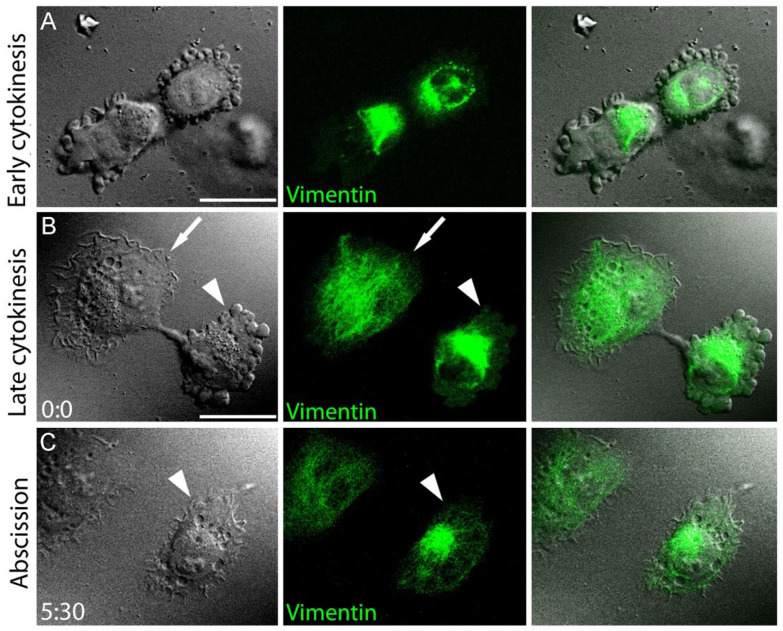
Subcellular distribution of VIFs correlates with blebbing behavior during mitosis. Mitotic EGFP-vimentin-expressing HT1080 cells shown by DIC (**left column**) and fluorescence microscopy (**middle column**), and by DIC/EGFP overlay (**right column**). (**A**) A cell at an early cytokinesis stage. VIFs (green) are enriched at the cleavage furrow and the nuclei, while blebbing occurs at the polar regions depleted of VIFs. (**B**,**C**) A mitotic cell shown during late cytokinesis (**B**) and after abscission (**C**). Time in min:sec. The daughter cell with broadly distributed VIFs (arrow) has lamellipodia, whereas the other daughter cell with perinuclear accumulation of VIFs (arrowhead) exhibits blebbing at late cytokinesis (**B**), but stops blebbing as soon as its VIF network is redistributed toward the periphery after abscission (**C**). Scale bars, 20 µm.

**Figure 5 cells-13-00105-f005:**
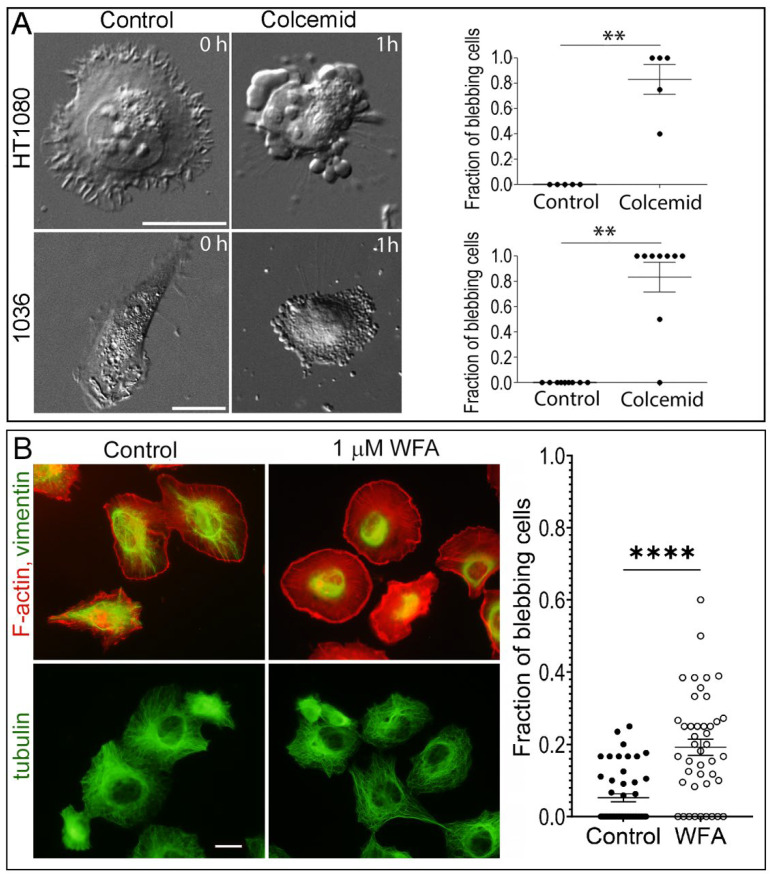
Pharmacological displacement of VIFs from cell periphery promotes blebbing. (**A**) Treatment of HT1080 (**top**) and 1036 (**bottom**) cells with 0.2 µM colcemid induces blebbing even without CK-666 treatment. Live-cell imaging by DIC microscopy before (**left**) and 1 h after colcemid treatment. Graphs show fractions of blebbing cells in individual experiments with and without colcemid treatment. Each data point represents an independent experiment (n = 71 HT1080 cells and 49 1036 cells across all experiments with 5–9 cells per experiment); lines show mean ± SEM; **, *p* < 0.0079, Mann–Whitney test. (**B**) Treatment of HT1080 cells with 1 μM WFA for 3 h. Immunofluorescence staining with antibodies to vimentin or α-tubulin and phalloidin. Scale bars, 20 μm. Graph shows fractions of blebbing cells per field of view in control (matching amount of DMSO) and WFA-treated cells (n = 547 control cells from 49 fields of view and 566 WFA-treated cells from 43 fields of view from 3 independent experiments); lines show mean ± SEM; ****, *p* < 0.0001, Mann–Whitney test.

**Figure 6 cells-13-00105-f006:**
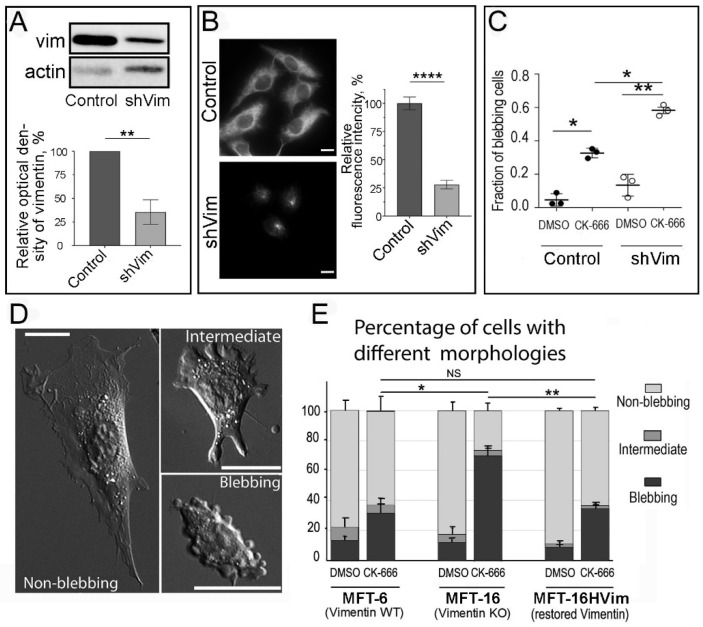
Genetic depletion of VIFs promotes blebbing. (**A**–**C**) Blebbing in HT1080scramble cells stably expressing scrambled shRNA (control) and vimentin-silenced HT1080shVim cells (shVim). (**A**) Representative Western blot of vimentin in control and HT1080shVim cells (**top**) and quantification of vimentin knockdown efficiency (bottom, % of control, mean ± SEM; n = 3 lysates/group; *p* < 0.05, *t*-test). Actin serves as loading control. (**B**) Immunofluorescence staining of vimentin (**left**) and relative immunofluorescence intensities of vimentin (right; mean ± SEM, n = 30 for each culture, ****, *p* < 0.0001, *t*-test) in control and HT1080shVim cells. Expression of shVim decreased the abundance of VIFs, which remained only in the perinuclear area. (**C**) Fractions of blebbing cells in control or HT1080shVim populations after treatment with 200 µM CK-666 or a matching amount of DMSO (mean +/− SEM from 3 independent experiments; n = 262, 312, and 397 for individual replicates). Mean values from each replicate were pooled and analyzed using Holm–Sidak’s multiple comparisons test, *, *p* < 0.05; **, *p* < 0.01. (**D**,**E**) Vimentin-null MFT-16 mouse fibroblasts exhibit greater fractions of blebbing cells after treatment with 100 µM CK-666, as compared with wild-type MFT-6 and vimentin-reconstituted MFT-16HVim fibroblasts. (**D**) Examples of mesenchymal, intermediate, and blebbing cell phenotypes. Scale bars, 10 µm. (**E**) Percentage of cells with different morphologies in the indicated conditions. Data show mean ± SEM for blebbing cells from three experiments; n = 65–203 cells per condition in each experiment. Statistical significance was determined using paired *t*-test; NS, not significant difference (*p* > 0.05); *, *p* < 0.05; **, *p* < 0.01.

## Data Availability

The data presented in this study are available in the article and Appendix A.

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
