# Peer review of "Plasma Membrane Blebbing Is Controlled by Subcellular Distribution of Vimentin Intermediate Filaments"

_cells, 2024, doi:10.3390/cells13010105_

Round 1
Reviewer 1 Report
Comments and Suggestions for Authors
The manuscript by Chikina et al. revealed an inverse relationship between membrane blebbing and localization of vimentin intermediate filaments at the cell edge in some cells of fibroblast origin. The finding is interesting as bleb-based cell motility is utilized by some tumors during dissemination. The manuscript is, in general, very well written and mainly accurately describes the findings. However, there are some questions to be addressed:
1) Figure 1:
a) Authors report that for blebbing (Fig.1A-C) and non-blebbing (Fig.1D-F) cells, they “did not detect striking differences in the distribution of microtubules or NMII”. However, visually, there is much higher density of the labeled NMII filaments (highlighted in blue) in the non-blebbing cell (F) compared to the blebbing cell (C). How did authors come to this conclusion about NMII filaments?
b) Next, by a naked untrained eye, it is hardly possible to distinguish 25-nm microtubules from 14-nm intermediate filaments on Fig. 1C and F. The authors should provide the images identical to C and F but with differentially highlighted microtubules and IFs.
c) What methods were used to quantify the filaments, how many cells were analyzed?
2) Result section 3.6. “Enforced removal of VIFs from the cell periphery promotes blebbing”
Sadly, authors used only one least appropriate approach to “remove VIFs from cell periphery”, namely depolymerization of microtubules with colcemid. This treatment removes not only VIFs but also all microtubules from the cell periphery, which makes it impossible to draw an accurate conclusion. Other approaches could be used here, such as treatment with 4-hydroxynonenal or prostaglandin A1 (both induce vimentin redistribution from the cell edge to the nucleus); ectopic expression of HIV-type I protease (which cleaves vimentin tail leading to abnormal filament distribution) or expression of tailless vimentin (such as in the Ref.#65 Duarte et al.). None of these have been attempted, however, and, therefore, at the minimum, the authors should tone down their interpretation of the results and wording of the title of this section (as colcemid removed both, VIFs and MTs).
Reviewer 2 Report
Comments and Suggestions for Authors
This is an interesting study showing that localization of vimentin intermediate filaments (VIFs) in fibroblasts affects cell membrane blebbing with increased blebbing occuring in regions depleted of VIFs. This finding is shown using multiple imaging techniques and using both pharmacologic and genetic depletion of vimentin. An understanding of how blebbing occurs in cells is potentially of high clinical significance given that blebbing has been found in immune cells and that some metastatic cells switch to blebbing prior to long distance migration.
While it is very clear from the data shown that perinuclear accumulation of VIFs promotes blebbing in fibroblasts whereas VIFs that extend to the cell periphery inhibit blebbing, there are a few significant concerns:
1) How blebbing is measured in all the experiments described in this paper needs to be clearly defined.
2) It is unclear why knockdown of vimentin or complete absence of vimentin in fibroblasts does not lead to increased cell blebbing at baseline (Figure 5E, F).
3) While describing a mechanism for how VIFs prevent blebbing may be beyond the scope of this paper, it would be nice to see some other functional effects of increased blebbing due to perinuclear localization of VIFs besides just cytokinesis. Does perinuclear localization of VIFs alter cell motility in any way?
4) In Figure 5A, were only 5- 9 cells for HT1080 cells and only 1-3 cells for 1036 cells examined per experiment? If so, why such a low number per experiment and how can the fraction of cells graphs in panel A be representative?
